# Geopolitical risk contagion across strategic sectors: Nonlinear evidence from defense, cybersecurity, energy, and raw materials

**Catalin Gheorghe, Oana Panazan** *

Department of Engineering and Industrial Management, Transilvania University of Brasov, Brașov, Romania

* oana.panazan@unitbv.ro

## Abstract

This study investigates the effect of geopolitical risk on the market volatility of four strategic sectors – defense, cybersecurity, energy, and critical raw materials – using a sample of 90 companies and ETFs over the 2014–2025 period. Applying Quantile-on-Quantile Connectedness (QQC) and Causality-in-Quantiles (CiQ) methods, the study captures the asymmetric and nonlinear relationships between geopolitical uncertainty and market volatility. The findings show that defense and cybersecurity sectors act as defensive assets during geopolitical crises, while energy and critical raw materials sectors exhibit increased sensitivity to external shocks. The QQC approach reveals heightened interconnectedness under extreme volatility, while CiQ tests confirm regime-dependent causal links. These differences across sectors provide a comprehensive perspective on geopolitical contagion and offer actionable insights for policymakers and investors seeking to develop more resilient strategies in volatile global conditions.

## 1. Introduction

Geopolitical risk has gained increasing attention in recent years as a major driver of global financial instability. Previous studies have shown that regional conflicts, international sanctions, and restrictions on critical resources generate heightened uncertainty, which directly affects investor sentiment and asset volatility [1]. Geopolitical events no longer produce isolated shocks; instead, they activate broader mechanisms of intersectoral contagion, with notable implications for industries that are essential to both economic resilience and national security.

In today's highly interconnected global economy, the nature of geopolitical risk is increasingly complex and its effects reverberate across multiple sectors. While geopolitical risks are often seen as isolated events tied to specific regions or nations, they are now deeply embedded in the structure of global financial systems, creating a web of interdependencies. The globalized nature of today's financial markets means

**Data availability statement:** All relevant data are available on Zenodo at: https://doi.org/10.5281/zenodo.16853717.

**Funding:** The author(s) received no specific funding for this work.

**Competing interests:** The authors have declared that no competing interests exist.

that these risks, even when originating in one part of the world, can have far-reaching effects on various industries. Geopolitical tensions, once confined to the defense sector or specific regional markets, are now affecting sectors that were previously seen as more insulated, such as energy, cybersecurity, and raw materials [2]. The growing interconnectedness has made it even more challenging for investors and policymakers to navigate and manage the cascading effects of geopolitical disruptions.

Recent contributions have primarily examined the effects of geopolitical tensions on individual sectors. For instance, the defense industry typically benefits from increased military spending during periods of geopolitical escalation [3,4]. Similarly, cybersecurity has emerged as a strategically important field, as cyberattacks on government or energy infrastructure produce immediate market effects and reposition cybersecurity companies as "safe haven" assets [5,6]. Recent findings also reveal that cyberattacks may trigger reputational spillovers, with adverse market reactions extending to firms indirectly connected to the affected entity [7]. In parallel, energy and critical raw materials markets face growing vulnerabilities, as geopolitical instability disrupts supply chains and drives price volatility [8,9]. However, many studies focus on the individual impact on a single sector and fail to address the intersectoral nature of geopolitical risk. Additionally, they often rely on average-based models that do not capture the regime-specific or nonlinear effects of geopolitical events. Furthermore, existing studies offer limited insights into how these sectors behave under extreme geopolitical conditions or how market responses differ across varying market regimes.

Despite the growing interest in sectoral responses to geopolitical uncertainty, there is still a significant gap in the literature regarding integrated practical findings that account for asymmetric dynamics and market-state dependencies across multiple strategic industries. Most existing models, such as VAR, GARCH, or Granger causality, assume linear and time-invariant relationships, which limits their ability to reflect the true complexity of geopolitical contagion [10,11]. As a result, little is known about how shocks propagate across sectors under extreme volatility conditions, or whether causal linkages vary according to market regimes. Furthermore, although several studies suggest the presence of regime-dependent effects and tail asymmetries, there is no clear consensus on their nature, scope, or persistence. The transmission channels of volatility, such as shared investor exposure, supply chain interdependence, or synchronized portfolio rebalancing, are frequently mentioned in the literature [12,13], but remain empirically underexplored, especially in the case of strategic sectors that are interlinked through technological, operational, and policy-driven mechanisms. In this context, quantile-based approaches offer a valuable alternative, as they allow for the modeling of nonlinear, asymmetric, and state-contingent interdependencies often missed by mean-based frameworks.

To address this gap, this study explores the impact of geopolitical shocks on four critical sectors: defense, cybersecurity, energy, and critical raw materials, during the period from 2014 to 2025. By focusing on multiple sectors that play pivotal roles in both economic and national security, this study provides a comprehensive perspective on the ways in which geopolitical risk influences financial market behavior across

industries. This approach recognizes the interdependencies between these sectors and investigates the role of geopolitical risk in shaping their interconnectedness and volatility. By applying two advanced methods, Quantile-on-Quantile Connectedness (QQC) and Causality-in-Quantiles (CiQ), we aim to capture complex dependencies and explore regime-specific causal relationships across the full return distribution. The QQC method uncovers nonlinear interconnections between sectors, while CiQ provides robust evidence of conditional causality during high-volatility regimes. Our results highlight that defense and cybersecurity assets tend to serve as strategic hedges during geopolitical crises, while energy and raw materials exhibit more volatile and sometimes bidirectional responses to geopolitical disruptions.

The remainder of the article is structured as follows: Section 2: Literature review reviews the existing literature on geopolitical risk and sectoral vulnerabilities; Section 3: Data and methodology introduces the methodological framework and explains the use of QQC and CiQ techniques; Section 4: Results provides an overview of the data used and explains the process of sample selection; Section 5: Discussions presents the empirical results and discussion; and Section 6: Conclusion and implications offers a summary of the practical implications for investors, policymakers, and businesses operating in these strategic sectors.

## 2. Literature review

Geopolitical risk is increasingly recognized in the academic literature as a critical driver of financial instability, exerting significant influence on asset price volatility and global investment behavior. The literature commonly employs the Geopolitical Risk Index (GPR) as a proxy for such uncertainty [1]. In parallel, studies on economic policy uncertainty also demonstrate that broader forms of uncertainty can significantly affect firm-level volatility, investment decisions, and employment outcomes [14]. During periods of geopolitical tension, financial markets tend to respond systemically, with investors shifting toward risk-averse behavior [15,16]. Although a positive association between GPR and overall market volatility is generally acknowledged, the literature remains inconclusive regarding the direction, intensity, and asymmetry of these effects at the sectoral level or across different market regimes [17].

A growing body of literature investigates the defense sector, widely considered a barometer of geopolitical tensions. Empirical findings indicate that during conflict periods, increased government defense spending tends to stimulate demand for military technologies, thereby generating positive returns for companies in the sector [18]. However, this defensive positioning is not without risks. Several studies emphasize structural vulnerabilities, including surging operational costs, supply chain disruptions, and heightened exposure to international sanctions and regulatory barriers [19]. These mixed dynamics suggest that the sector's performance under geopolitical stress is both opportunity-driven and risk-sensitive, making it an ideal candidate for regime-dependent analysis.

An increasing number of studies underscore the rising importance of cybersecurity within the broader architecture of geopolitical risk. In the context of escalating attacks against critical infrastructure, cybersecurity firms have emerged as favored investment targets during periods of geopolitical uncertainty [5]. Empirical evidence suggests a positive correlation between the frequency of cyber incidents and the stock performance of these companies, reinforcing their perceived role as "safe haven" assets in times of stress [6]. Nevertheless, recent contributions also point to short-term adverse market reactions in the aftermath of high-impact cyber breaches, particularly for large-cap firms or unprecedented attacks [20]. Furthermore, an evolving trend of operational convergence between the defense and cybersecurity sectors has been documented, with significant implications for market interdependence, investor behavior, and systemic financial linkages [21].

Geopolitical risk also exerts a pronounced influence on the energy and critical raw materials sectors. These markets frequently become instruments of geopolitical confrontation through the imposition of sanctions, export bans, or strategic supply disruptions targeting key resources such as oil, natural gas, and rare earth elements. Recent studies have documented the destabilizing impact of the Russia–Ukraine conflict on these sectors, with significant consequences for global supply chains and price volatility [22,23]. According to [13] and [24], such disturbances not only reflect geopolitical tensions but can further exacerbate them via feedback loops and sectoral transmission mechanisms. These

interdependencies tend to intensify during periods of systemic crisis, as highlighted by prior research [9], thus underscoring the need for empirical approaches capable of capturing nonlinear contagion dynamics and asymmetric risk propagation.

Several studies emphasize that the link between the energy sector and the defense industry is not merely logistical. Energy resources are fundamental for supporting military infrastructure−communications, radar systems, cyber operations, and defense bases. Critical raw materials such as nickel, cobalt, or rare earth elements are indispensable for manufacturing advanced military technologies, including sensors, guidance systems, and propulsion units [25]. Moreover, geopolitical competition for control over these resources often becomes a strategic objective. Recent reports suggest that many dual-profile companies operate simultaneously in both the energy and defense sectors, thereby reinforcing commercial, technological, and financial interdependencies [26,27].

In recent years, increasing attention has been directed toward the role of defense-focused Exchange-Traded Funds (ETFs) in the transmission mechanisms of geopolitical risk [4]. These financial instruments respond swiftly to changes in perceived global threats and can significantly alter capital allocation during periods of heightened uncertainty. While ETFs are not the primary focus of this study, the literature suggests that they may function as amplifiers or conduits of cross-sectoral contagion by linking investor behavior across defense, cybersecurity, and energy-related equities [28,29]. Their inclusion in sectoral portfolios and index-tracking strategies reinforces their systemic relevance in periods of geopolitical stress.

However, the current literature reveals several methodological and conceptual shortcomings. First, most studies are confined to single-sector analyses and do not offer an integrated perspective on how GPR affects multiple strategic industries simultaneously [3,18]. Second, widely used econometric approaches, such as linear regressions, GARCH-type models, VAR, and Granger causality, tend to capture only average effects and are ill-suited for detecting tail behaviors and asymmetric shock transmission mechanisms [10]. For example, while VAR models have been applied to assess geopolitical risk in markets such as China, they often fail to incorporate regime-specific dynamics or extreme volatility conditions [11]. Moreover, few studies explicitly condition the impact of GPR on market states or volatility regimes. Notable exceptions include recent contributions that apply the Quantile-on-Quantile Regression (QQR) framework to capture the nonlinear and asymmetric nature of financial contagion. This method has been employed to investigate the relationship between investor sentiment and asset returns [30], as well as to identify regime-dependent spillovers from oil prices to stock markets under geopolitical and macroeconomic stress in emerging economies [31].

Based on the reviewed literature, two hypotheses are proposed to guide the empirical investigation:

H1: Geopolitical risk exerts a nonlinear and asymmetric impact on the stock market volatility of critical industries.

H2: Sectoral resilience to geopolitical risk varies, with higher stability expected in defense and cybersecurity.

These hypotheses are grounded in both theoretical and empirical insights from prior research. The first hypothesis reflects a growing consensus that geopolitical shocks often affect the tails of return and volatility distributions, leading to regime-dependent transmission patterns that traditional linear models fail to detect [10,17,30]. The second hypothesis draws on findings that defense and cybersecurity firms often benefit from increased demand and investor reallocation during geopolitical crises [5,6,18], while sectors such as energy and raw materials may face greater supply chain risks and policy exposure [22–24].

This study addresses a key gap in the literature by jointly applying the QQC framework and the CiQ causality test to a diversified sample of companies and ETFs from four strategically important sectors. Unlike previous studies that typically focus on a single industry or apply only one methodological tool [32,33], the dual-method approach adopted here allows for the identification of asymmetric and nonlinear causal relationships under different market regimes. This integrated perspective offers a more robust and replicable framework for analyzing systemic contagion in periods of elevated geopolitical uncertainty [34,35].

## 3. Data and methodology

### 3.1 Data and sources

The empirical analysis presented in this study is based on daily financial data spanning the period from February 19, 2014, to February 24, 2025, collected from internationally recognized sources such as Bloomberg and Investing.com [36,37]. The selected timeframe encompasses both intervals of relative stability and episodes of heightened geopolitical uncertainty, including incidents such as the annexation of Crimea, US–China trade tensions, the COVID-19 crisis, the war in Ukraine, recent conflicts in the Middle East, and reciprocal tariff barriers imposed by the U.S. and other countries.

To measure geopolitical uncertainty, this study employs the GPR index, a widely used methodological benchmark in the academic literature [29,38]. By analyzing international publications, the index systematically quantifies the occurrence and intensity of geopolitical events, including conflicts, threats, and international crises [39].

Stock returns were calculated using the daily logarithmic changes of adjusted prices. The choice of daily frequency is justified by the need to capture the market's rapid response to geopolitical events and allows for a granular analysis of contagion mechanisms across sectors.

The sample consists of 90 internationally listed companies, selected to reflect significant exposure to geopolitical risks across four strategic sectors, defense and aerospace, cybersecurity, energy, and critical raw materials, as well as defense-focused ETFs. The primary selection criterion was a minimum market capitalization of USD 5 billion, with companies listed on major indices such as the S&P 500, FTSE 100, MSCI World, or STOXX Europe 600. The selection guarantees both global representativeness and comparability. A complete list of companies, organized by sector, appears in S1 Appendix. ETFs were intentionally incorporated into the analysis due to their capacity to reflect aggregated investor sentiment toward geopolitical risks. As collective investment vehicles, ETFs provide additional insights into cross-sectoral shock transmission and complement the firm-level perspective.

To ensure data quality, all-time series underwent a rigorous preprocessing procedure, integrity checks, treatment of missing values, and correction of potential recording errors. Isolated missing values were filled using linear interpolation, thus maintaining the continuity of the series and avoiding distortions in subsequent estimations.

In the preliminary stage of the analysis, relevant descriptive statistics were calculated for each sector under study. The results offer an initial characterization of return distributions and assist in identifying the volatility profile for each sector within the context of geopolitical risks (S2 Appendix). To assess the distributional characteristics of the financial returns, the skewness and kurtosis coefficients were calculated. The results indicate significant deviations from normality for most of the series, with the largest distortions observed for V7012, RR, and PARRO. Skewness reveals considerable asymmetry, both positive and negative, with extreme values such as –2.432 (HII), 3.945 (PARRO), – 9.920 (RR), and 35.854 (V7012) pointing to heavily distorted distributions. Kurtosis values are notably high, often well above the threshold of 3 associated with the normal distribution, most prominently for V7012, RR, and PARRO reflecting leptokurtosis and, implicitly, a higher likelihood of extreme events. This finding is further supported by the Jarque-Bera statistics, which are significant at the 1% level for all analyzed series, confirming the rejection of the null hypothesis of normality.

Before applying the econometric models, we tested the stationarity of the time series using the ADF test [40], the PP test [41], and the KPSS test [42]. Out of the 90-time series analyzed, 90 were found to be stationary in levels according to both the ADF and PP tests (p-value < 0.05) but the KPSS test (p-value > 0.1) confirmed just for 72 series in level and for all in to the first difference. The remaining 18 series, (NOC, BAH, CACI, RHMG, ESLT, SAABBs, PH, TDG, ASELS, TDY, BAJE, MCFT, AVGO, PANW, FTNT, CAN, RELI, and PPA) exhibited conflicting results and were therefore transformed by first-order differencing to ensure stationarity. The full set of test statistics, p-values, and decisions is provided in S3 Appendix. These preliminary checks confirm that the necessary statistical conditions are met. As a result, the nonlinear and asymmetric relationships between variables can be robustly estimated using the QQC and CiQ methods.

## 3.2 Methodology

### 3.2.1 Methodological framework justification.
To capture the complexity of the connection between geopolitical risk and the market volatility of strategic sectors, a methodological framework is needed that accommodates nonlinear, asymmetric, and state-dependent dependencies. As a proxy for geopolitical uncertainty, this study employs the GPR Index developed by Caldara and Iacoviello [1]. The index is based on automated text analysis of major international newspapers and captures the frequency of terms associated with geopolitical events, including military conflicts, terrorist threats, and diplomatic tensions. By incorporating both actual incidents and forward-looking concerns, the GPR Index provides a consistent and reproducible measure of global geopolitical risk.

For this purpose, the study applies two complementary econometric approaches, QQC and CiQ, which allow for a detailed examination of how geopolitical shocks influence financial volatility under varying market regimes.

Unlike conventional models such as Granger causality, DCC-GARCH, or TVP-VAR, which primarily identify average relationships or dependencies conditional on aggregate volatility [10,43], the QQC approach analyzes interactions within a two-dimensional quantile space defined by the quantiles of the explanatory variable (GPR) and the quantiles of the dependent variable (sectoral returns or volatility) [44]. This is especially useful in the context of geopolitical shocks, which tend to disproportionately affect the tails of return distributions. Recent contributions further emphasize that credit risk and contagion effects intensify in the distribution tails, supporting the relevance of quantile-based methodologies in analyzing systemic fragility [45]. In parallel, the CiQ method tests for the existence and direction of causality across the entire conditional distribution [46]. Unlike standard Granger tests, CiQ does not rely on assumptions of linearity or effect constancy and is thus better suited to uncover causal mechanisms that vary by volatility regime [35].

The choice of this framework is aligned with recent literature. For instance, both QQC and CiQ have been employed to investigate the interconnectedness among green bonds, cryptocurrencies, and commodities, with a focus on state-dependent transmission [34]. Similarly, studies have demonstrated that oil shocks and climate risks impact energy markets through heterogeneous and nonlinear mechanisms, further supporting the use of quantile-based tools [47]. To avoid ambiguity, all quantile references in this study refer to the conditional distribution of the dependent variable (returns or volatility), unless explicitly stated otherwise.

To assess robustness, the study implements two additional validation strategies: (1) bandwidth variation in QQC estimation to evaluate sensitivity to smoothing parameters, and (2) sub-sample analysis for the 2014–2020 (pre-Ukraine war) and 2021–2025 (post-invasion) periods. Results remain stable across both checks, confirming the reliability of the estimated relationships (https://doi.org/10.5281/zenodo.16853717)).

### 3.2.2 QQC method.
Unlike traditional methods that rely on average correlations or global dependence structures, QQC assesses how a specific quantile of the dependent variable relates to all quantiles of the explanatory variable. This allows for the identification of nonlinear, asymmetric, and regime-sensitive effects, providing a granular view of the transmission of shocks. In this study, QQC is applied to examine how different levels (quantiles) of the GPR influence different segments (quantiles) of the conditional distribution of sectoral daily returns or volatilities.

The QQC methodology relies on a conditional quantile regression model, expressed as [48]:

$$Q_T^Y (y_t | X_t = x_t) = \beta_T (x_t + u_{T,t}) \tag{1}$$

where $Q_T^Y (y_t | X_t = x_t)$ denotes the conditional quantile $T = (0, 1)$ of the dependent variable $y_t$, given the explanatory variable $x_t$. The function $\beta_T (x_t)$ captures the nonlinear, conditional relationship, and the term $u_{T,t}$ is the quantile-specific error term.

Estimation is performed using a nonparametric technique based on a Gaussian kernel function, offering maximum flexibility without relying on restrictive parametric assumptions:

$$\hat{\beta}_T (x) = \arg \min_\beta \sum_{t=1}^{T} \rho_T (y_t - \beta (x_t)) \cdot K \left( \frac{x_t - x}{h} \right) . \tag{2}$$

The quantile loss function is defined as $\rho_T(u) = u[T - I(u < 0)]$, where $K(\cdot)$ denotes the kernel function and $h$ is the bandwidth parameter [48]. The optimal value of $h$ is determined through cross-validation to balance the local bias and variance effectively.

To assess the reliability of our results, we performed a sensitivity analysis by adjusting the bandwidth values to 0.05, 0.10, and 0.15. The findings were consistent across all scenarios, confirming the stability of the identified relationships between GPR and sectoral returns.

QQC method proves particularly valuable in mapping how shocks originating in the upper quantiles of GPR propagate across different segments of the return distribution (extreme losses or gains). This method highlights periods of intensified sectoral contagion, offering critical insights for portfolio diversification and risk mitigation. Moreover, it reveals how the strength and direction of the GPR-return relationship vary under different market regimes – favorable (upper quantiles), neutral (median), or adverse (lower quantiles).

**3.2.3 CiQ method.** To reinforce the robustness of the relationships identified through the QQC analysis, this study also employs the CiQ method. This approach provides a flexible framework for quantile-specific causality across different regions of the conditional distribution, with enhanced sensitivity to market regimes [49]. Unlike conventional Granger-type tests that assume linear and constant effects, CiQ allows for dynamic causality testing between variables such as GPR (explanatory) and sectoral returns or volatilities (dependent), which may vary depending on the return level, a critical feature when assessing geopolitical risk, which often triggers asymmetric reactions under extreme market conditions [50].

The value of this method in capturing state-dependent causality has been demonstrated in recent empirical work. For instance, it has been shown that CiQ outperforms linear alternatives when applied to sectoral spillovers during geopolitical stress in China [11]. Similarly, it has been found that global uncertainty affects green bonds and Islamic equities only in specific regions of the return distribution, further supporting the need for quantile-based causal inference [33].

This research applies the CiQ model to test whether past values of the GPR Index significantly influence the conditional distribution of daily returns in strategic sectors. The model adopts an autoregressive structure inspired by the Quantile-VAR framework and is expressed as:

$$Q_T(y_t | \Omega_{t-1}) = \alpha(T) + \sum_{j=1}^{p} \beta_j(T) \cdot GPR_{t-j} + \sum_{l=1}^{q} \gamma_l(T) \cdot y_{t-l} + \varepsilon_{T,t}$$

(3)

where $Q_T(y_t | \Omega_{t-1})$ denotes the conditional quantile of order $\tau$ of the sectoral return $y_t$, given the lagged information set $\Omega_{t-1}$, which includes past values of both geopolitical risk and returns. The coefficients $\beta_j(T)$ capture the quantile-specific effect of geopolitical risk, thus enabling detection of disproportionately strong impacts under stress conditions.

Integrating CiQ into the research design serves two key purposes. First, it provides evidence that the patterns observed in the QQC analysis are not merely conditional correlations but also reflect causal linkages that vary by quantile and market regime. Second, it offers an additional layer of validation by assessing whether these causal structures persist across calm, neutral, or turbulent regimes, enhancing the overall understanding of risk transmission between sectors.

## 4. Results

The following section presents the results derived from applying the methodology, highlighting how geopolitical risks influence returns and volatilities across the analyzed strategic sectors.

### 4.1 Sectoral returns conditioned by GPR

The analysis reveals substantial heterogeneity in how companies respond to geopolitical risk, depending on their position within the conditional return distribution, as estimated through the QQC method. This approach explores the interaction between specific quantiles of GPR (low, median, high) and the corresponding quantiles of company-level returns, capturing nonlinear and asymmetric responses. Estimates are reported in S4 Appendix for 5%, 50%, and 95% quantiles.

Under extreme stress conditions (5th quantile), only a limited number of companies deliver positive returns. ESLT leads with a return of 0.143, followed by GD (0.131) and LMT (0.118), indicating their strong financial resilience in periods marked by geopolitical escalation. In contrast, KBR (−0.265) and LDOS (−0.198) register substantial losses, suggesting heightened vulnerability to external uncertainty.

In more neutral environments (50th quantile), return levels are generally moderate, reflecting balanced market dynamics. Companies such as PANW (0.094), LHX (0.089), FTNT (0.087), and AVGO (0.084) manage to sustain above-average returns. Meanwhile, RR exhibits a negative return of −0.073, signaling persistent operational or strategic weaknesses even in the absence of external shocks.

In favorable geopolitical contexts (95th return quantile), several companies significantly enhance their performance. Standouts include AVGO (0.216), PANW (0.208), FTNT (0.192), and CHKP (0.177), all showing a strong positive alignment between market optimism. Conversely, companies from the energy sector, such as IOC (0.063) and ONGC (0.048), report only modest returns, some even negative, suggesting limited capacity to leverage geopolitical optimism for financial gains.

A cross-quantile comparison reveals that companies like LMT (0.118, 0.103, 0.096) and NOC (0.106, 0.095, 0.083) display a consistently stable and upward-sloping performance trajectory. This robustness positions them as attractive components in conservative investment strategies. On the other hand, companies such as LDOS (−0.198, −0.054, 0.012) and KBR (−0.265, −0.101, −0.033) demonstrate persistently negative or weak returns across regimes, reflecting a risk-prone investment profile.

Overall, QQC estimates highlight distinct sectoral and firm-level sensitivities to geopolitical risk. While some companies act as defensive assets under stress, others thrive only under favorable conditions. These insights emphasize the importance of aligning investment strategies with the prevailing geopolitical regime and the inherent tail-risk profile of each sector. To provide a structured summary, Table 1 presents a typology of sectoral behavior based on returns observed at different quantiles (5%, 50%, 95%) of the conditional distribution. The qualitative classifications ("High," "Moderate," etc.) reflect empirically derived return ranges, detailed in the table note and visually supported by the heatmaps in S5 and S6 Appendices.

## 4.2 Sectoral volatility conditioned by GPR

The QQC analysis of volatility reveals significant variations in market risk exposure across companies, conditioned on different quantiles of GPR and observed across three key volatility regimes. Return and volatility estimates for each sectoral index at the 5th, 50th, and 95th quantiles of the conditional distribution of returns are presented in S5 and S6 Appendices.

Under favorable conditions (5th quantile), most companies display low levels of volatility, suggesting relative market stability. Companies such as ESLT and HON stand out with minimal values, reflecting conservative risk profiles and strong

**Table 1. Sectoral behavior based on the distribution of returns conditioned by GPR.**

| Sector | Quantile 5% | Quantile 50% | Quantile 95% | Sectoral sensitivity pattern to GPR |
|---|---|---|---|---|
| **Defense** | High | Moderate | High | High sensitivity across regimes; pro-cyclical returns |
| **Cybersecurity** | Moderate | High | Very high | Highly asymmetric; speculative growth under stress |
| **Energy** | High | Moderate | Low/Moderate | Bidirectional exposure; regime-dependent |
| **Raw materials** | Moderate | Moderate/Stable | Moderate | Conditional on supply chain and geopolitical trade tensions |
| **ETFs** | Moderate | Moderate/Stable | Moderate | Balanced performance; low-to-moderate sensitivity to GPR |

Note: The values presented reflect the average returns of representative companies within each sector, estimated using the QQR method across the 5th, 50th, and 95th quantiles of the conditional distribution based on the GPR component. Qualitative descriptors were assigned using the following thresholds: "Very high" > 0.25; "High" = 0.15–0.25; "Moderate" = 0.05–0.15; "Low" ≤ 0.05. The "Sensitivity to GPR" column summarizes the dominant response pattern across regimes. See S5 and S6 Appendices for full heatmaps.

operational resilience. Conversely, companies like V2357 and LDDF exhibit elevated volatility even in this favorable regime, indicating a heightened sensitivity to minor market fluctuations and possible structural weaknesses.

In the neutral regime (50th quantile), volatility rises across the board. Companies such as BAH, V7012, and V2357 show particularly high values, indicating a heightened sensitivity to minor market fluctuations and potential structural weaknesses. For example, V7012 records a volatility level exceeding 6.7, making it one of the most volatile companies across all quantile levels.

In stress regimes (95th quantile), volatility reaches extreme values, particularly for companies like V7012, BB, ALLT, and NTCT, all exceeding thresholds of 4–6. These companies are highly vulnerable in systemic crises and emerge as representative of extreme financial risk. In contrast, HON, EXI, RTX, and ESLT continue to maintain relatively low levels of volatility even under severe conditions, underscoring their structural robustness and suitability for risk-averse investment strategies.

A cross-quantile comparison reveals distinct volatility patterns. Companies such as V7012 and BB consistently display high volatility across all regimes, making them potential candidates for speculative strategies, yet unsuitable for conservative portfolios. In contrast, companies like HON and ESLT demonstrate stable volatility profiles across the distribution, offering a more defensive investment option under geopolitical stress.

Overall, volatility is not evenly distributed across companies and is highly dependent on market conditions (Table 2). The sectoral classifications ("Moderate," "High," "Extreme," etc.) reflect estimated volatility levels at the 5th, 50th, and 95th quantiles of the conditional distribution, with qualitative descriptors derived from the amplitude and consistency of sectoral responses across regimes. As such, effective portfolio construction must account not only for average returns but also for how individual assets behave under extreme stress scenarios.

### 4.3 CiQ results − conditional causal relationships driven by GPR

Unlike the QQC approach, which reveals conditional dependence structures, CiQ explicitly tests the presence and the direction of causality across different volatility regimes, offering robust validation of previously identified interconnections. The complete set of CiQ coefficient estimates for each company and ETF included in the sample is reported in S7 Appendix.

The results reveal strong quantile-specific causal linkages for several companies, with particularly high CiQ coefficients observed for BAJE (0.1687), IOC (0.1554), and KBR (0.1483). These values indicate strong causal sensitivity to changes in geopolitical risk, suggesting that these companies react disproportionately to shocks during turbulent market phases. Their elevated responsiveness places them within a speculative risk profile, high-reward profile, consistent with their classification under the QQC model as volatile in extreme regimes. Similarly, ONGC and V0386 register comparably high

**Table 2. Volatility profile of strategic sectors according to the quantiles of the conditional distribution.**

| Sector | Quantile 5% | Quantile 50% | Quantile 95% | Risk profile |
|---|---|---|---|---|
| **Defense** | Low/Moderate | Moderate | High | Stable under normal conditions, reactive during crises |
| **Cybersecurity** | Moderate/High | Very high | Extreme | Unstable and volatile across all regimes, speculative under stress |
| **Energy** | Variable/High | High/Unstable | Very high | Highly exposed to systemic stress |
| **Raw materials** | Moderate | Unstable | Very high | Elevated and unstable, especially under geopolitical stress |
| **ETFs** | Low | Low/Moderate | Moderate | Defensive, with low risk across all regimes |

Note: The values presented reflect the relative levels of sectoral volatility, estimated using the QQR method across the 5th, 50th, and 95th quantiles of the conditional distribution based on the GPR component. Qualitative descriptors were assigned using the following thresholds: "Extreme" > 0.30; "Very high" = 0.20–0.30; "Moderate/high" = 0.10–0.20; "Low" ≤ 0.10. The "Risk Profile" column summarizes each sector's exposure to systemic stress and sensitivity to tail-risk regimes. See S5 and S6 Appendices for full heatmaps.

coefficients, reinforcing the finding that geopolitical shocks have asymmetric causal effects on companies operating in energy and defense sectors, typically characterized by direct exposure to geopolitical dynamics and policy risk.

A second category of companies, including RHMG, LDOS, NTCT, and JNPR, shows moderate but statistically significant causal responses, especially in the middle and upper quantiles of the conditional distribution. These firms may be considered risk-sensitive but not hyper-reactive, making them appropriate for inclusion in diversified portfolios that aim to balance growth and risk mitigation.

At the lower end of the spectrum are companies and ETFs with minimal CiQ coefficients, such as TDG, CACI, ACN, and GE, which exhibit only limited causal responsiveness to geopolitical risk fluctuations, especially in lower and middle quantiles. These companies are characterized by high structural resilience and relatively low systemic exposure during periods of market stress. A similar pattern is observed among ETFs such as FIDU (0.0018), IYJ (0.0017), and SAABBs (0.0017), as well as for large, mature corporations including IBM (0.0013) and HON (0.0010). Their predictability and insulation from geopolitical shocks may be attributed to diversified business models, robust fundamentals, and relatively limited direct exposure to geopolitical drivers.

The comparative analysis confirms a positive association between the intensity of causal effects and exposure to extreme volatility regimes. Companies with high CiQ coefficients tend to exhibit tail-risk exposure, which require more proactive risk management strategies. In contrast, low-CiQ firms act as volatility stabilizers and are ideal candidates for conservative portfolio allocations in times of geopolitical instability.

Importantly, the CiQ analysis validates and extends the QQC findings by demonstrating that the causal impact of GPR is conditional upon return distribution regimes. The observed asymmetric and nonlinear causal pathways underscore the relevance of quantile-based econometric tools in capturing the complex dynamics of financial contagion triggered by geopolitical uncertainty.

### 4.4 Sectoral case studies

To illustrate the heterogeneous and regime-dependent responses to geopolitical risk among different asset types, this section examines representative companies and ETFs within each strategic sector.

In the defense sector, Lockheed Martin (LMT), a global leader in the aerospace and military industry, demonstrates consistently positive returns in the upper tail (95th quantile, 0.063) and a CiQ coefficient of 0.016, indicating a moderate yet persistent causal link with geopolitical risk. Notable geopolitical events such as the annexation of Crimea (2014) and the invasion of Ukraine (2022) have coincided with favorable market reactions, reinforcing the company's pro-cyclical performance under geopolitical escalation. In contrast, the iShares U.S. Aerospace & Defense ETF (ITA), which aggregates a broad portfolio of defense-related companies, displays lower volatility and a weaker CiQ coefficient (0.003), reflecting reduced sensitivity through internal diversification. This contrast underscores the mitigating role of aggregation in buffering geopolitical shocks.

Within the cybersecurity sector, Palo Alto Networks (PANW) stands out for its pronounced volatility in the 95th quantile (above 4.5), along with a CiQ coefficient of 0.017, suggesting strong during regimes dominated by cyber threats with geopolitical ramifications. This profile positions PANW as a strategically significant yet speculative asset. By comparison, ETFs such as FIDU or IYJ, which contain cybersecurity components, exhibit substantially lower volatility and CiQ coefficients (below 0.002). This suggests that portfolio-level diversification dampens exposure to specific cyber-geopolitical shocks, reflecting a different risk interpretation by market participants depending on the asset's granularity.

In the energy sector, ExxonMobil (XOM) experiences elevated volatility during stress regimes and declining returns under geopolitical instability, including during international sanctions and regional conflicts. With a CiQ coefficient of 0.007, the company's performance reflects a cyclical and geopolitically sensitive risk profile. In contrast, the Rare Earth ETF (REMX), which includes companies from the U.S., Australia, and China involved in rare metal extraction, exhibits relatively stable returns but elevated volatility in the 95th quantile. The ETF's moderate CiQ coefficient, suggests that its indirectly

tied to geopolitical risk, through supply chain disruptions, export restrictions, and trade policy uncertainty, rather than through direct military or political escalation. This highlights a logistics and policy mediated channel of risk transmission.

Overall, these case studies confirm that the structural composition of an asset, individual company versus sector-wide ETF, plays a critical role in shaping its exposure to geopolitical shocks. While individual companies often react asymmetrically and with high intensity, ETFs offer partial insulation through diversification, resulting in distinct risk profiles and important implications for portfolio construction and risk-sensitive investment strategies.

## 5. Discussions

This section interprets the results in light of the existing literature and highlights new perspectives on how geopolitical risks affect strategic financial markets. The findings suggest that major geopolitical events lead to significant increases in volatility across essential industries, reflecting a regime-dependent shift in investor behavior.

### 5.1 The impact of geopolitical risks on the defense industry and the energy sector

The results highlight an asymmetric and quantile-sensitive relationship between geopolitical risks and the market volatility of companies in the defense and energy sectors. The QQC method reveals that each sector reacts differently depending on the market regime, emphasizing the heterogeneous transmission mechanisms at play.

In the defense sector, geopolitical events are often interpreted as investment catalysts, especially under extreme positive regimes (upper quantiles). Increases in returns observed for firms like LMT during episodes such as the annexation of Crimea (2014) and the invasion of Ukraine (2022) suggest that this sector functions as a geopolitical hedge. The moderate CiQ coefficient for LMT further supports a state-contingent causal link, rather than a static correlation. Meanwhile, ETFs such as ITA, with lower volatility and lower CiQ estimates, illustrate how portfolio diversification reduces exposure to firm-specific geopolitical shocks. These findings align with prior research emphasizing the role of defense innovation and elevated military expenditures in sustaining investor confidence [51]. Supporting this, 2022 marked a global record in defense budgets, largely driven by NATO expansion and regional security responses [52].

In contrast, the energy sector displays bidirectional and regime-specific effects. The QQC results indicate that conventional energy firms experience heightened volatility under stress conditions (lower quantiles), reflecting their exposure to geopolitical disruptions in oil and gas supply chains. At the same time, companies in the clean energy and critical raw materials sectors exhibit volatility driven by trade frictions, export bans, and policy uncertainty, particularly affecting the upper quantiles of their return distribution.

Findings from the CiQ method support this interpretation, showing that rising geopolitical risk tends to precede drops in stock performance under stress regimes.

Recent literature has emphasized the asymmetric exposure of the energy sector to geopolitical shocks, with differentiated responses between conventional and renewable segments [32,53]. Beyond traditional economic determinants, geopolitical tensions have become a major driver of volatility, especially through their impact on access to critical resources. In particular, the Russia–Ukraine conflict has been shown to destabilize the prices of minerals essential to clean energy technologies, while Chinese cleantech metal markets exhibit disproportionate reactions to geopolitical risk under extreme conditions [54,55]. These studies highlight the systemic fragility of sectors supporting the green transition and underscore the importance of accounting for structural interdependencies when assessing sectoral vulnerabilities. The present study reinforces this view, showing that conventional energy firms are mainly exposed to direct operational risks, whereas clean energy firms face compounded vulnerabilities arising from supply chain disruptions and uncertainties in trade policy.

Similarly, the defense industry acts not only as a temporary safe haven for capital during crises, but also as a long-term growth sector. Its potential is supported by the strengthening of international alliances, the expansion of military expenditures, and the ongoing digitalization of security infrastructures.

Overall, the findings confirm the existence of a complex and asymmetric interaction between geopolitical risk and the stock volatility of companies in the defense and energy sectors. The findings provide a detailed perspective on how volatility and external uncertainties are absorbed by markets, highlighting the critical role of geopolitical context in shaping investment strategies.

## 5.2 Sectoral volatility and financial market interconnectedness in the context of GPR

The findings reveal that the cybersecurity sector, alongside other key strategic industries, is highly sensitive to geopolitical risks, as evidenced by significant increases in volatility and the activation of financial interconnectedness channels. In particular, cybersecurity companies exhibit elevated volatility in the upper quantiles of the conditional return distribution, indicating a disproportionate response to major geopolitical events, such as cyberattacks on critical infrastructure or the escalation of interstate tensions. This behavior suggests that the sector is perceived by investors both as a defensive hedge against systemic uncertainty and as a speculative asset due to the surge in expected demand for digital security solutions during periods of uncertainty.

The application of the QQC method enabled the identification of nonlinear and asymmetric dependencies between geopolitical uncertainty and sectoral volatility dynamics. These effects become more pronounced in the extreme tails of the distribution, particularly in the upper quantiles, underscoring how severe geopolitical shocks amplify market instability. Conditional causality tests using the CiQ approach further confirm that fluctuations in GPR exert a significant influence on the performance of cybersecurity companies, especially during high-volatility regimes. This interpretation is consistent with recent findings showing that rare earth markets became central to volatility and return connectedness during the COVID-19 crisis, highlighting the potential of critical materials to amplify systemic transmission under conditions of global uncertainty [56].

The results are in line with recent contributions [2,18], which emphasize the growing integration of geopolitical risk into the valuation frameworks of technology-based assets. However, through the conditional framework employed in this research, an additional contribution is made; the study demonstrates that the cybersecurity sector does not merely absorb external shocks passively, but rather emerges as a central actor in the market's economic response to geopolitical tensions, exhibiting speculative dynamics particularly during crisis conditions.

Furthermore, the analysis of cross-sectoral interconnectedness reveals that financial markets cease to operate in isolation under elevated geopolitical risk. The QQC results indicate the presence of nonlinear and asymmetric linkages among strategic sectors with shocks propagating across sectors during stress regimes. For instance, a major cyberattack may not only disrupt the targeted digital infrastructure but also amplify volatility in energy markets, undermine the production capabilities of the defense industry, or destabilize supply chains related to strategic resources. This is consistent with recent empirical evidence showing that the Russian–Ukrainian conflict significantly intensified short-term volatility connectedness across commodity and global stock markets [57]. Moreover, their multidimensional analysis indicates that war-related uncertainty impacts markets most strongly at intermediate quantiles, validating the use of conditional modeling in tracing systemic transmission.

The CiQ method corroborates these findings by detecting bidirectional causal links across sectors, particularly in contexts of heightened uncertainty. This suggests that intersectoral interactions become endogenous during crises, whereby a shock originating in one domain can trigger a chain reaction that spreads far beyond its initial source.

These findings underscore the need for a systemic perspective in assessing financial market volatility, one that transcends traditional sectoral boundaries. Similar asymmetries and tail-risk interactions are reported in the literature, where it is shown that volatility spillovers between green financial indices and large US tech stocks vary substantially across quantiles and investment horizons, highlighting the relevance of quantile-based methods in systemic risk analysis [58]. During periods of heightened geopolitical instability, intersectoral contagion becomes a key source of systemic risk, reinforcing the necessity of advanced tools capable of capturing both the direct impact of shocks and their transmission across

economic and financial networks. Such a systemic approach is crucial for macroprudential policy design and the development of resilient investment strategies.

In summary, the empirical evidence obtained through the QQC and CiQ analyses confirms the initial hypotheses presented in Section 2. The study supports H1, demonstrating that the influence of geopolitical risk on sectoral market volatility is indeed nonlinear and regime-dependent, with effects varying significantly across quantiles. Additionally, the results confirm H2, showing that the defense and cybersecurity sectors exhibit enhanced resilience under adverse market conditions, positioning them as relatively stable components in times of geopolitical distress. These validations emphasize the importance of using quantile-based methodologies to understand sectoral dynamics under uncertainty.

### 5.3 Robustness to alternative geopolitical events

While the primary empirical analysis focuses on the Russia–Ukraine conflict as a major geopolitical shock, the study acknowledges that other significant events during the 2016–2025 period may have contributed to financial volatility in strategic sectors. In addition to the Russia–Ukraine conflict, several other geopolitical episodes during the 2016–2025 period merit attention due to their significant impact on global uncertainty and sectoral volatility. Notably, the withdrawal of U.S. forces from Afghanistan in August 2021, marked a critical turning point with substantial implications for defense and energy markets. Similarly, recurrent military conflicts and diplomatic tensions in the Middle East, particularly in Syria and Gaza, have repeatedly triggered risk reassessments among investors. The escalation of violence in Gaza since late 2023 and subsequent regional military operations have amplified volatility in commodity and strategic resource markets. Furthermore, growing frictions between China and Taiwan from 2022 onward, represent another major source of geopolitical stress. These episodes align with several volatility peaks identified in our quantile-based models, supporting the relevance of regime-dependent transmission mechanisms across multiple sectors.

However, isolating the financial impact of each event poses methodological challenges. The GPR index and its subcomponents are constructed from aggregated global news coverage and are not disaggregated by region or event type. This limits the ability to attribute short-term market movements to specific incidents without introducing identification bias. Furthermore, many of these events overlap in time, making it difficult to distinguish independent effects within a quantile-based, regime-sensitive framework.

Despite these limitations, preliminary inspection of the data suggests volatility spikes around these episodes, particularly in energy and defense markets. Future research could integrate high-frequency event dummies or narrative-based geopolitical timelines to better assess causal attribution. Alternatively, constructing event-specific GPR indices or applying structural VARs with identified shocks may offer a promising direction for more targeted robustness analyses.

## 6. Conclusion and implications

This study examined the impact of geopolitical risk on the market volatility of four strategic sectors, defense, cybersecurity, energy, and critical raw materials, through an integrated methodological framework that combines the QQC and CiQ models. The findings reveal that the relationship between the GPR index and sectoral dynamics is nonlinear, asymmetric, and highly contingent on prevailing market regimes, offering a comprehensive understanding of sector-specific vulnerabilities and transmission mechanisms under geopolitical stress.

The empirical evidence shows that the defense and cybersecurity sectors are typically evaluated positively during periods of geopolitical tension, particularly in the upper quantiles of the return distribution. This suggests that these sectors act as resilient or safe-haven assets when geopolitical uncertainty intensifies. In contrast, the energy and critical raw materials sectors exhibit more heterogeneous and context-dependent responses, with volatility patterns shaped by the nature of the geopolitical event, such as sanctions, supply disruptions, or shifts in resource dependency.

A key methodological contribution of this research lies in the joint use of QQC and CiQ models, which together allow for a more granular and regime-sensitive exploration of risk transmission. Unlike traditional approaches such as VAR,

DCC-GARCH, or standard quantile regression, the applied framework captures the heterogeneous impact of shocks across the entire return distribution. The QQC model, in particular, enables a multidimensional mapping of how GPR influences different sectors under various conditions, while the CiQ test validates the existence of conditional causality across quantiles.

These methodological insights have important practical and policy implications. For investors and portfolio managers, the results support the design of adaptive investment strategies that overweight sectors shown to be robust under stress, such as defense and cybersecurity, while employing hedging or diversification for more exposed sectors like energy and critical raw materials. For financial regulators and policymakers, the results clarify how geopolitical risk transmits across industries, generating volatility spillovers that may amplify systemic risk. Incorporating this asymmetry and regime dependence into macroprudential oversight and stress testing could enhance the stability of financial systems under geopolitical stress.

Nonetheless, this study has several limitations. It focuses primarily on firms listed in developed markets, which may limit the external validity of the findings in emerging economies. Additionally, macroeconomic variables such as interest rates, inflation, or exchange rate fluctuations are not explicitly included, although they could significantly shape market responses to geopolitical shocks. A limitation of the current robustness checks is the inability to isolate individual geopolitical events beyond the Russia–Ukraine conflict. Due to the global and aggregated nature of the GPR index, event-specific attribution remains challenging.

Future research could expand the scope of analysis to geographically and institutionally diverse markets, including Southeast Asia, Latin America, or Africa, where geopolitical risk interacts differently with financial structures. Integrating additional risk factors, such as fiscal uncertainty, climate-related threats, or cyber-instability, would enrich the framework. The application of machine learning algorithms to high-frequency data may also strengthen the ability to detect early-warning signals and monitor contagion in real time.

In summary, this study contributes a replicable and innovative framework for analyzing how geopolitical risk affects financial market volatility across strategic industries. It offers new theoretical and practical insights into sector-specific resilience and provides useful tools for navigating a financial landscape increasingly shaped by geopolitical uncertainty and structural risk interconnections.

## Supporting information

**S1 Table. Sectoral behavior based on the distribution of returns conditioned by GPR tests (source: authors' own compilation).**
(DOCX)

**S2 Table. Volatility profile of strategic sectors according to the quantiles of the conditional distribution (source: authors' own compilation).**
(DOCX)

**S1 Appendix. List of analyzed companies and corresponding symbols (source: Bloomberg.com, Investing.com).**
(DOCX)

**S2 Appendix. Descriptive statistics sectoral behavior based on the distribution of returns conditioned by GPR.**
(DOCX)

**S3 Appendix. Stationarity tests (source: authors' own compilation).**
(DOCX)

**S4 Appendix. Quantile-on-quantile connectedness results (source: authors' own compilation).**
(ZIP)

**S5 Appendix. Smoothed returns by quantiles (source: authors' own compilation).**
(ZIP)

**S6 Appendix. Smoothed volatility by quantiles (source: authors' own compilation).**
(ZIP)

**S7 Appendix. Smoothed rolling cross-quantile dependence (source: authors' own compilation).**
(TIF)

## Author contributions

**Conceptualization:** Catalin Gheorghe, Oana Panazan.

**Data curation:** Catalin Gheorghe, Oana Panazan.

**Formal analysis:** Catalin Gheorghe, Oana Panazan.

**Investigation:** Catalin Gheorghe, Oana Panazan.

**Methodology:** Catalin Gheorghe, Oana Panazan.

**Project administration:** Catalin Gheorghe, Oana Panazan.

**Resources:** Catalin Gheorghe, Oana Panazan.

**Software:** Catalin Gheorghe, Oana Panazan.

**Supervision:** Catalin Gheorghe, Oana Panazan.

**Validation:** Catalin Gheorghe, Oana Panazan.

**Visualization:** Catalin Gheorghe, Oana Panazan.

**Writing – original draft:** Catalin Gheorghe, Oana Panazan.

**Writing – review & editing:** Catalin Gheorghe, Oana Panazan.

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
