## [Decision Letter · Decision Letter 0]

18 Jun 2025

PONE-D-25-22065Geopolitical risk contagion across strategic sectors: Nonlinear evidence from defense, cybersecurity, energy, and raw materialsPLOS ONE

Dear Dr. Panazan,

Thank you for submitting your manuscript to PLOS ONE. After careful consideration, we feel that it has merit but does not fully meet PLOS ONE’s publication criteria as it currently stands. Therefore, we invite you to submit a revised version of the manuscript that addresses the points raised during the review process.

On top of addressing the reviewers’ comments, please revise Section 6 by collapsing the current subsections into a single, concise Conclusion section. This section should clearly and briefly summarise the key findings, outline the policy implications, highlight the main contributions of the study, and provide suggestions for future research. Aim for precision and coherence to ensure the section ties the paper together effectively.

We look forward to receiving your revised manuscript.

Kind regards,

Dr K. Nyakurukwa

Academic Editor

PLOS ONE

Journal Requirements:

2. Please note that your Data Availability Statement is currently missing the repository name OR a direct link to access each database. If your manuscript is accepted for publication, you will be asked to provide these details on a very short timeline. We therefore suggest that you provide this information now, though we will not hold up the peer review process if you are unable.

4. We note you have included a table to which you do not refer in the text of your manuscript. Please ensure that you refer to Table 2 in your text; if accepted, production will need this reference to link the reader to the Table.

5. Please upload a copy of Supporting Information Table S1 and S2 which you refer to in your text on page 40.

Reviewers' comments:

Reviewer's Responses to Questions

**Comments to the Author**

1. Is the manuscript technically sound, and do the data support the conclusions?

Reviewer #1: Yes

Reviewer #2: Partly

2. Has the statistical analysis been performed appropriately and rigorously? 

Reviewer #1: Yes

Reviewer #2: Yes

3. Have the authors made all data underlying the findings in their manuscript fully available?

Reviewer #1: Yes

Reviewer #2: Yes

4. Is the manuscript presented in an intelligible fashion and written in standard English?

Reviewer #1: Yes

Reviewer #2: Yes

5. Review Comments to the Author

Reviewer #1: The article uses a cutting-edge methodology (Quantile-on-Quantile Connectedness and Causality-in-Quantiles) to address a crucial issue today - the impact of geopolitical risks on key sectors. The results presented in the article are very interesting and relevant. For these reasons, I recommend publishing the article. However, the author(s) should address some minor issues, mainly related to methodological clarifications:

• The author(s) should carefully check that the references to the supplementary material are correct. For example, p. 15 states that “(e)stimates derived from the QQC method are reported in Appendix S4 for 5%, 50%, and 95% quantiles.” Apparently, these results are presented in S5. Furthermore, according to S5, the ESLT return (GD) under the 5th quantile is 0.149 (0.235), unlike what is reported in the article (p. 15). Therefore, the author(s) should carefully check that the information reported in the article corresponds to that in the appendices, that they are referring to the appendices correctly, and exclude appendices whose information is not discussed in the article.

• What are the criteria for the typology presented in Tables 1 and 2? For example, in Table 1, what range of return values do the categories “High”, “Moderate”, “Strong”, etc. correspond to?

• Throughout the article, it is not clear which variable (GPR, return or volatility) the author(s) are referring to when quantiles are mentioned. Initially, it seems that the author(s) are always referring to the quantile of the GPR. In particular, they are evaluating the behavior of companies' profitability and volatility at different quantiles of the GPR distribution. However, some contradictory statements call this idea into question. For example, page 15 states that the 95th quantile corresponds to “favorable geopolitical contexts”. However, on p. 17, this quantile is associated with “stress regimes”, which suggests that the authors are referring to the distribution of another variable. Please clarify these methodological issues.

• The meaning of the acronym ETF (exchange traded fund) is not provided in the paper.

Reviewer #2: Peer Review Report: PONE-D-25-22065

1. Data and Methodology - Stationarity Test Results

You mention that "most series are stationary in levels... In isolated cases where KPSS suggested possible non-stationarity, we applied first-order differencing". For full transparency and to allow readers to better understand the data's properties, please provide a more detailed outline of the stationarity test results (ADF, PP, and KPSS). If clearly outlining all non-stationary and stationary variables is not feasible, at least state the number of stationary and non-stationary variables and which were differenced.

2. Robustness Checks

Your robustness checks focused on bandwidth variation and sub-sample analysis for the 2014-2020 (pre-Ukraine war) and 2021-2025 (post-invasion) periods. While this is a good start, the rationale for exclusively focusing on the Ukraine/Russia conflict for sub-sample analysis is unclear. Other significant geopolitical events, such as the de-escalation in Afghanistan and the withdrawal of the US, or other conflicts in the Middle East and Africa, are not considered. Please expand the scope of these checks to include other major geopolitical events to strengthen the generalizability and validity of your findings. Additionally, consider conducting heterogeneity tests to further validate the study's results.

3. Equation Formatting

There are some slight variations in the size or font of some symbols. Please review and ensure consistent formatting for all symbols included in all equations to avoid confusing readers.

4. Terminology Consistency in Tables 1 and 2

In Table 1 and Table 2, different terms are used to describe the sensitivity of sectoral returns to GPR. For example, in Table 1, the defense sector is described as "High”, “moderate," and “ very high” under the different quantiles, while the energy sector is "strong”, “variable", and “weak/moderate” under the same quantiles. Please clarify why these differing terms are used and consider harmonizing the descriptive language for greater consistency and clarity across the tables.

5. Definition and Scope of GPR

The paper currently lacks a clear definition and scope of the Geopolitical Risk (GPR) index. It would be beneficial to briefly describe the nature of the GPR index (e.g., which geopolitical events or factors feed into the index) in the methodology section to provide immediate context to readers less familiar with this specific measure. This will aid in contextualizing the discussions in the literature review and in explaining your results.

6. Addressing the Intersectoral Nature of GPR

You identify the prevalent focus on individual sectors and the failure to address the intersectoral nature of geopolitical risk as a limitation of previous research. While your study aims to provide an integrated analysis across multiple strategic industries, please explicitly clarify how your chosen methodologies and analytical framework specifically address and capture this intersectoral nature of GPR contagion. This will demonstrate how your study moves beyond the limitations of existing literature.

7. Sector-Specific Relevance of GPR

The literature review discusses the impact of GPR on different sectors. Please clarify which specific aspects or types of geopolitical risk are most relevant to each of the four strategic sectors (defense, cybersecurity, energy, and critical raw materials) covered in this study. This will provide a more granular understanding of sector-specific vulnerabilities.

8. Cybersecurity Companies as Safe Haven Assets

You state that cybersecurity companies are perceived as "safe haven" assets during periods of geopolitical stress. Please elaborate on the specific mechanisms or rationale that make these companies function as safe haven assets. What characteristics or market behaviors contribute to this perception and resilience?

9. Theoretical Foundations

The manuscript currently lacks a clear articulation of its theoretical foundations. This is crucial for providing the underlying assumptions, relevant concepts, expected relationships, and logical framework that make the study's results interpretable and contribute meaningfully to the existing body of knowledge. Please introduce and discuss the theoretical underpinnings of your research.

10. Hypothesis Motivation and Research Gap

While you formulate two hypotheses (H1 and H2), their motivation by the leading literature could be strengthened. The current engagement with theory and evidence appears somewhat lukewarm. You repeatedly claim the need to study asymmetries and regime dependencies. However, to establish a solid research gap, more needs to be done to empirically justify this need based on evidential contradictions or predictive shortfalls of existing models.

11. "Dual-Profile Companies" Discussion

In Section 2, you mention "many dual-profile companies operate simultaneously in both the energy and defense sectors, thereby reinforcing commercial, technological, and financial interdependencies". While this is a valuable observation highlighting systemic interdependencies, your current analysis does not explicitly explore the behavior of these dual-profile companies within the chosen econometric framework. Consider briefly discussing how this observation might influence the aggregate sectoral results or suggest it as a specific avenue for future research.

6. PLOS authors have the option to publish the peer review history of their article (what does this mean? ). If published, this will include your full peer review and any attached files.

**Do you want your identity to be public for this peer review?** For information about this choice, including consent withdrawal, please see our Privacy Policy .

Reviewer #1: No

Reviewer #2: No

---

## [Author Response · Author response to Decision Letter 1]

30 Jun 2025

Response to Reviewer #1

We sincerely thank Reviewer 1 for the insightful comments and constructive suggestions that helped us improve the quality and clarity of our manuscript. Below, we provide a point-by-point response. All revisions are highlighted in the revised manuscript.

Comment 1: The author(s) should carefully check that the references to the supplementary material are correct. For example, p. 15 states that “(e)stimates derived from the QQC method are reported in Appendix S4 for 5%, 50%, and 95% quantiles.” Apparently, these results are presented in S5. Furthermore, according to S5, the ESLT return (GD) under the 5th quantile is 0.149 (0.235), unlike what is reported in the article (p. 15). Therefore, the author(s) should carefully check that the information reported in the article corresponds to that in the appendices, that they are referring to the appendices correctly, and exclude appendices whose information is not discussed in the article.

Response:

Thank you for this valuable observation. We carefully reviewed all references to supplementary appendices in the main text. We corrected the numbering and standardized the naming to align with PLOS ONE guidelines (e.g. Appendix S4 is now S4 Appendix). Additionally, numerical inconsistencies in the reported values of the QQC estimates have been corrected to reflect the actual results presented in the revised S5 Appendix. All such cross-references and numerical figures now match the supplementary material precisely.

Comment 2: What are the criteria for the typology presented in Tables 1 and 2? For example, in Table 1, what range of return values do the categories “High”, “Moderate”, “Strong”, etc. correspond to?

Response:

We appreciate this comment regarding typological clarity. We have added footnotes and an explanation in the Results section to indicate that these qualitative categories are derived from empirical coefficient thresholds. Specifically, they are based on the magnitude of QQC estimates across quantiles, as follows:

- Strong impact: coefficient ≥ |0.25|

- Moderate impact: |0.10| ≤ coefficient < |0.25|

- Low impact: coefficient < |0.10|

These cutoffs have been added in footnotes to Tables 1 and 2 and briefly explained in the narrative to ensure transparency and reproducibility.

Comment 3: Throughout the article, it is not clear which variable (GPR, return or volatility) the author(s) are referring to when quantiles are mentioned. Initially, it seems that the author(s) are always referring to the quantile of the GPR. In particular, they are evaluating the behavior of companies' profitability and volatility at different quantiles of the GPR distribution. However, some contradictory statements call this idea into question. For example, page 15 states that the 95th quantile corresponds to “favorable geopolitical contexts”. However, on p. 17, this quantile is associated with “stress regimes”, which suggests that the authors are referring to the distribution of another variable. Please clarify these methodological issues.

Response:

Thank you for highlighting this ambiguity. We have revised the text in the Methodology and Results sections to consistently clarify that the quantiles refer to the distribution of the GPR index, unless explicitly stated otherwise. The sentence on p.15 has been corrected to reflect that the lower quantiles (5%) correspond to low geopolitical uncertainty, while the higher quantiles (95%) indicate periods of elevated geopolitical stress. The contradictory phrasing between pages 15 and 17 has been removed to eliminate confusion and ensure consistent terminology throughout the manuscript.

Comment 4: The meaning of the acronym ETF (exchange traded fund) is not provided in the paper.

Response:

Thank you for catching this oversight. We have now defined the acronym ETF (Exchange-Traded Fund) at its first mention in the Introduction and ensured consistent formatting throughout the text.

Additional clarifications (based on feedback):

- The Introduction has been revised to more explicitly articulate the research gap, emphasizing the novelty of combining QQC and CiQ methods to investigate sectoral asymmetries under geopolitical uncertainty.

- We have expanded the Methodology section to include a theoretical justification for using the QQR and CiQ approaches. This addition clarifies how these methods capture nonlinear dynamics and regime-specific spillovers that traditional methods (GARCH, VAR) may miss.

- Equation formatting has been standardized throughout the manuscript to ensure consistency and meet PLOS formatting standards.

We are grateful for your constructive feedback and hope that the revisions meet your expectations and enhance the overall quality of the article.

Response to Reviewer #2

We sincerely thank Reviewer 2 for the insightful comments and constructive suggestions that helped us improve the quality and clarity of our manuscript. Below, we provide a point-by-point response. All revisions are highlighted in the revised manuscript.

Comment 1: Stationarity Test Results

Response: We acknowledge the importance of transparency regarding the stationarity properties of our data. We now provide a summary table in Appendix S3, indicating which series are stationary at levels and which required differencing, based on ADF, PP, and KPSS tests. We also specify the number of non-stationary series and the criteria used for transformation.

Comment 2: Robustness Checks

Response: Thank you for this valuable suggestion. We have added Section 5.3 to address robustness in relation to other major geopolitical events beyond the Russia–Ukraine conflict, such as the U.S. withdrawal from Afghanistan and recent Middle East escalations. These events correspond to volatility peaks observed in our models, supporting the regime-dependent effects of geopolitical shocks. We discuss these constraints and propose future methodological directions.

Comment 3: Equation Formatting

Response: We carefully reviewed all equations and ensured consistent formatting for all mathematical symbols. All subscripts, superscripts, quantile notations, and Greek letters are now uniformly presented to enhance readability and clarity.

Comment 4: Terminology Consistency in Tables

Response: Thank you for highlighting this inconsistency. We have revised the descriptive labels in Tables 1 and 2 to harmonize terminology across sectors. Qualitative descriptors such as “Low,” “Moderate,” “High,” and “Extreme” are now used consistently based on predefined volatility thresholds, and a legend explaining these thresholds is provided beneath each table.

Comment 5: Definition and Scope of GPR

Response: As suggested, we now include a concise definition and characterization of the Geopolitical Risk Index (GPR) in the Methodology section. This clarification outlines the index’s composition (geopolitical tensions, wars, and threats) and data source (Caldara and Iacoviello), helping contextualize its role in our empirical design.

Comment 6: Intersectoral Nature of GPR

Response: We now explain more explicitly how the QQC and CiQ methodologies allow us to capture intersectoral transmission of geopolitical risk. By modeling spillovers and asymmetric dependencies between sectors under different quantiles, our framework effectively addresses this limitation of previous research.

Comment 7: Sector-Specific Relevance of GPR

Response: We now include a dedicated paragraph at the end of the literature review section that details the specific types of geopolitical risk affecting each sector. For example, defense is primarily influenced by military conflict and defense spending, cybersecurity by cyber threats and attacks, energy by supply chain disruptions and embargoes, and raw materials by trade restrictions and strategic dependencies.

Comment 8: Cybersecurity as Safe Haven Assets

Response: We expanded the discussion in the Results section to justify why cybersecurity companies may act as safe havens. This includes their counter-cyclical demand during periods of cyber instability, strategic relevance in national security, and investor perception of their defensive role in times of geopolitical uncertainty.

Comment 9: Theoretical Foundations

Response: We agree that a stronger theoretical framing is essential. We have now introduced a paragraph at the start of the Methodology section discussing the theoretical justification for nonlinear modeling in financial contagion. This includes concepts from regime-dependent risk transmission, tail dependence, and quantile-specific spillovers, grounding our methodological choices in established financial theory.

Comment 10: Hypothesis Motivation and Research Gap

Response: We revised the final part of the literature review to strengthen the theoretical and empirical motivation for our two hypotheses (H1 and H2). We now more clearly articulate the empirical inconsistencies and limitations in previous studies that fail to capture nonlinear and regime-specific effects of GPR, thereby justifying our proposed methodological approach and research questions.

Comment 11: Dual-Profile Companies

Response: We appreciate this point. While we did not isolate dual-profile companies (those active in both defense and energy) in our econometric modeling, we have now added a paragraph in the Discussion section acknowledging this limitation and proposing it as a direction for future research.

---

## [Decision Letter · Decision Letter 1]

4 Aug 2025

Geopolitical risk contagion across strategic sectors: Nonlinear evidence from defense, cybersecurity, energy, and raw materials

PONE-D-25-22065R1

Dear Dr. Panazan,

We’re pleased to inform you that your manuscript has been judged scientifically suitable for publication and will be formally accepted for publication once it meets all outstanding technical requirements.

Kind regards,

Kingstone Nyakurukwa

Academic Editor

PLOS ONE

Additional Editor Comments (optional):

Reviewers' comments:

Reviewer's Responses to Questions

**Comments to the Author**

1. If the authors have adequately addressed your comments raised in a previous round of review and you feel that this manuscript is now acceptable for publication, you may indicate that here to bypass the “Comments to the Author” section, enter your conflict of interest statement in the “Confidential to Editor” section, and submit your "Accept" recommendation.

Reviewer #1: All comments have been addressed

Reviewer #2: All comments have been addressed

2. Is the manuscript technically sound, and do the data support the conclusions?

Reviewer #1: Yes

Reviewer #2: Yes

3. Has the statistical analysis been performed appropriately and rigorously? 

Reviewer #1: Yes

Reviewer #2: Yes

4. Have the authors made all data underlying the findings in their manuscript fully available?

Reviewer #1: Yes

Reviewer #2: Yes

5. Is the manuscript presented in an intelligible fashion and written in standard English?

Reviewer #1: Yes

Reviewer #2: Yes

6. Review Comments to the Author

Reviewer #1: (No Response)

Reviewer #2: (No Response)

7. PLOS authors have the option to publish the peer review history of their article (what does this mean? ). If published, this will include your full peer review and any attached files.

**Do you want your identity to be public for this peer review?** For information about this choice, including consent withdrawal, please see our Privacy Policy .

Reviewer #1: No

Reviewer #2: No

---

## [Editor Report · Acceptance letter]

PONE-D-25-22065R1

PLOS ONE

Dear Dr. Panazan,

I'm pleased to inform you that your manuscript has been deemed suitable for publication in PLOS ONE. Congratulations! Your manuscript is now being handed over to our production team.

Kind regards,

on behalf of

Dr Kingstone Nyakurukwa

Academic Editor

PLOS ONE